# Enhancing the Accuracy of Land Cover Classification by Airborne LiDAR Data and WorldView-2 Satellite Imagery

Chun-Ta Wei [1], Ming-Da Tsai [2,*], Yu-Lung Chang [2] and Ming-Chih Jason Wang [3]

1   School of Defense Science, Chung Cheng Institute of Technology, National Defense University, Taoyuan 33551, Taiwan; 1120512301@ccit.ndu.edu.tw
2   Department of Environmental Information and Engineering, Chung Cheng Institute of Technology, National Defense University, Taoyuan 33551, Taiwan; chang72118@gmail.com
3   Department of History and Geography, University of Taipei, Taipei 100234, Taiwan; wmcsm@go.utaipei.edu.tw
*   Correspondence: tsai.md5478@ccit.ndu.edu.tw; Tel.: +886-3-380-0364 (ext. 307)

**Abstract:** The Full Waveform LiDAR system has been developed and used commercially all over the world. It acts to record the complete time of a laser pulse and has a high-resolution sampling interval compared to the traditional multiple-echo LiDAR, which only provides signals within a single target range. This study area mainly collects data from Riegl LMS-Q680i Full Waveform LiDAR and WorldView-2 satellite imagery, which focuses on buildings, vegetation, grassland, asphalt roads and other ground types as the surface objects. The amplitude and pulse width are selected as waveform basic parameters. The parameter of topography is slope, and the height classification parameters of the test ground are 0–0.5 m, 0.5–2.5 m, and 2.5 m. To eliminate noise, the neighborhood average is applied on the LiDAR parameter values and analyzed as the classification accuracy comparison. This survey uses Decision Tree as the classification method. Comparing the data between neighborhood average and non-neighborhood average, the data classification accuracy improves by 7%, and Kappa improves by 5.92%. NDVI image data are utilized to distinguish the artificial from natural ground. The results show that the neighborhood average with previous data can improve the classification accuracy by 5%, and Kappa improves by 4.25%. By adding NIR-2 of WorldView-2 satellite imagery to the neighborhood average analysis, the overall classification accuracy is improved by 2%, and the Kappa value by 1.21%. This article shows that utilizing the analysis of neighborhood average and image parameters can effectively improve the classification accuracy of land covers.

**Keywords:** LiDAR; full waveform; decision tree; accuracy

## 1. Introduction

This study aims to use WorldView-2 satellite imagery and LMS-Q680i airborne LiDAR data to produce research on classification and interpretation of features. In the section concerning satellite imagery, the majority of the image data are from the WorldView-2 satellite, which provides eight different wavebands. Each of these has different radiation characteristics on the same surface objects. The paper will use the 8th band, the near-infrared band-2 (NIR-2), to collect image data. Then, the 5th band, red and 7th band, and near-infrared band-1 (NIR-1) will be used to calculate the Normalized Difference Vegetation Index (NDVI) as a significant parameter of the image classification. Next, the decision tree is applied to analyze the LiDAR data classification parameters (echo width, amplitude, slope, custom height values, after neighborhood averages' echo width, amplitude, slope) to define the land covers, which could be called the "LiDAR decision tree process".

This survey is divided into two experimental stages. In the first stage, the LiDAR decision-tree process mentioned above is applied. In the second phase of the experiment, the poor classification results are chosen, and then the image NDVI and image NIR-2 band

parameters are added to verify whether the addition of LiDAR data to the image data can improve the overall classification accuracy.

Guo et al. [1] used the random forest classification method to classify urban landmarks by traditional multiple-echo, full-waveform LiDAR and multispectral image parameters. In their research, four parameters, which are elevation, amplitude, width, and backward reflection waveform, are applied to define the building class. The waveform parameters of the artificial ground class have relatively low importance.

To detect vegetation in urban areas, Höfle et al. [2] use a neural network, decision-tree classifier, and full waveform LiDAR as parameters to collect data. The effects of multiple echoes, gridded segmentation, radiometric correction, and classification on vegetation detection in urban areas are analyzed, compared, and discussed. The results show that the accuracy of classification reaches up to 98% with the above classification methods and point-cloud density can be effectively reduced to 10 (pts./m$^2$). Full waveform LiDAR data definitely enhance the characterization conditions for describing objects, and can effectively help to distinguish objects. Wu et al. [3] selected the Random Forest (RF) as a classifier. The spectral bands of GF-2, NDVI, and normalized digital surface model derived from LiDAR data, and their grey-level co-occurrence matrix (GLCM) textures including mean, variance, homogeneity, contrast, dissimilarity, entropy, second moment, and correlation, were generated to create seven scenarios with different combinations of RF input variables. This study aims to provide a reference for the efficient improvement of land-cover classification, and to offer support for extending the applications of classification algorithms and data sources.

## 2. Theory

### 2.1. LiDAR

The full waveform LiDAR system records the echo waveform by dense sampling, recording the continuous digital number (DN) value, recording the echo waveform reflected by different substances, and obtaining the point-cloud information in the full waveform through waveform post-processing. The point-cloud information of the full waveform is more complete than that of the traditional multi-echo point-cloud information [4]; the complete waveform information is recorded, making the point-cloud data denser. It is more valuable than the traditional multiple echo in the study of classification and application [5].

The full waveform LiDAR point cloud will affect the complexity of the return waveform due to the size of the footprint when recording the surface. The large footprint covers a large area of the ground. The waveform complexity recorded is relatively high, which is difficult to analyze and process in post-processing analysis. The advantage of a large footprint is that the research area can be quickly scanned. The small-footprint point cloud covers a small area and the waveform information recorded is relatively simple. It is necessary to increase the scanning frequency to achieve an increase in point-cloud density. At present, most commercial LiDARs are mainly small footprints [6].

### 2.2. NDVI

The more vigorous the growth of colorful plants, the more red light chlorophyll absorbs, and the stronger the reflection of infrared light. The difference between the two is used: proportion of subtraction and addition. NDVI has no unit, and its value ranges from −1.0 to 1.0. The larger the value, the greater the growth of the plant. The formula is as follows:

$$NDVI = \frac{(NIR - Red)}{(NIR + Red)} \tag{1}$$

In the formula, *NIR* and *Red* respectively represent the near infrared light band and the red-light band, and the output value of *NDVI* is between −1.0 and 1.0. A positive increase in the value of *NDVI* represents the degree of increase in vegetation. A value close to 0 or negative, indicates the absence of vegetation. For places with a vegetation distribution, there will be a higher *NDVI* value. With higher reflection of near-infrared light band and

lower reflection of red-light band, a place with dense vegetation distribution yields an NDVI value between 0.1 and 0.6, and its value depends on the density and greening at the top of the plant. Soil and rock materials will produce values close to 0, because the two have similar values in the red and near-infrared bands, and water, clouds, and snow will have higher red-band values than in the near-infrared band, so they will produce negative values. Therefore, to use the *NDVI* method to extract vegetation information, the threshold must be manually adjusted before it can be applied to different satellite images.

### 2.3. Decision Tree Classification

The decision-tree mining technology is an algorithm that divides and predicts the structure of the tree and the diagram. Multistage classification techniques are also possible, in which a series of decisions are taken to determine the correct label for a pixel. The more common multistage classifiers are called decision trees [7]. They consist of a number of connected classifiers (or decision nodes), none of which is expected to perform the complete segmentation of the image data set.

Frequently, decision tree strategies can be designed manually, particularly when they are required to perform quite specific labelling tasks [8]. However, as with single stage classifier and neural network training, it would be of value to have automated design procedures available. Since the number of possible tree structures, even for a moderately small number of classes, is astronomical, it is very difficult to design an optimal classifier [9]. Classification accuracy and efficiency, however, rely heavily on the tree chosen. Therefore, various heuristic methods for decision-tree design have been developed, details of which can be found in Safavian and Landgrebe [10].

To make the design task easier, binary decision trees are often adopted. Discrimination ability is not necessarily weakened by choosing a binary approach, since a general decision tree can be uniquely transformed into an equivalent binary tree [11]. The decision tree is also called the classification tree; the decision tree is one of the common functions of classification in the data mining processing technology. This study uses classification and regression trees (CART) in MATLAB to classify point-cloud data. CART is one of the commonly used data detection and data analysis tools, which can automatically detect the potential structure, important patterns, and relationships of highly complex data. The basic principle of CART is to use entropy calculation to divide the sample data into two parts, and each time the data will be divided into two subsets. Then, according to different training samples, classification is carried out layer by layer. The tree node at the top of the classification tree usually has the most influence on the classified data, and the influence of the classification tree node gradually decreases according to the layering [12].

### 2.4. Accuracy Evaluation

Olofsson et al. [13] discussed the basic protocols needed to produce scientifically rigorous and transparent accuracy and area estimates. This set of good practice recommendations provides guidelines to assist scientists and practitioners in designing and implementing precision assessment and area estimation methods for land change assessment using remote sensing.

For the evaluation and validation of the classification results, the accuracy of the classification results will be evaluated according to the real point-cloud data drawn manually. The method used to assess the classification accuracy is the error matrix. As shown in Table 1, this method uses geo-authentic references to verify the quality of classified data, further validated by user accuracy (UA), producer accuracy (PA), and overall accuracy (OA) [14,15].

It is assumed that the categories classified by the ground truth data are N categories from A to N in total. The decision-tree classification results used in this study are a to n in a total of n categories. They are listed on the abscissa and ordinate of the error matrix, respectively. The value of each coordinate from $X(1, 1)$ to $X(n, N)$ in the matrix is filled in with the number of classified point clouds.

**Table 1.** Error Matrix.

| Classification Results | Ground Truth | | | | | |
| --- | --- | --- | --- | --- | --- | --- |
| | Class A | Class B | ... | Class N | Total Category Pixels | PA |
| Class a | $X(1, 1)$ | $X(1, 2)$ | ... | $X(1, N)$ | $\sum_{i=1}^{N} X(1, i)$ | $\frac{X(1,1)}{\sum_{i=1}^{N} X(1,i)}$ |
| Class b | $X(2, 1)$ | $X(2, 2)$ | ... | $X(2, N)$ | $\sum_{i=1}^{N} X(2, i)$ | $\frac{X(2,2)}{\sum_{i=1}^{N} X(2,i)}$ |
| ⋮ | ⋮ | ⋮ | ⋮ | ⋮ | ⋮ | ⋮ |
| Class n | $X(n, 1)$ | $X(n, 2)$ | ... | $X(n, N)$ | $\sum_{i=1}^{N} X(n, i)$ | $\frac{X(n,N)}{\sum_{i=1}^{N} X(n,i)}$ |
| Total Ground truth Category | $\sum_{i=1}^{N} X(i, 1)$ | $\sum_{i=1}^{N} X(i, 2)$ | ... | $\sum_{i=1}^{N} X(i, N)$ | | |
| UA | $\frac{X(1,1)}{\sum_{i=1}^{N} X(i,1)}$ | $\frac{X(2,2)}{\sum_{i=1}^{N} X(i,2)}$ | ... | $\frac{X(n,N)}{\sum_{i=1}^{N} X(i,N)}$ | | |

User Accuracy (UA): The meaning of the row coordinates of Category A in the error matrix is the same as the method used in this study to classify Category A objects in the coordinates classified for real data in light. The number of point clouds in Category A is filled in $X(1, 1)$. In Category A coordinates, the number of pixels in Category B is filled in $X(2, 1)$. In order to determine if point clouds in class A are classified as N by this study, or if they are filled in sequentially, after completing the coordinates of class A row, it can be determined that in the coordinates of real data class A objects, classes B to n from $X(2, 1)$ to $X(n, 1)$ are the number of misclassified point clouds, and classes a from $X(1, 1)$ are the correct number of point clouds that can be completely classified into class A coordinates by this study. Therefore, user accuracy is defined as the true classified objects. This method can be used to classify the ratio to the true objects, and then the error matrix is completed.

$$UA = \frac{X(1,1)}{\sum_{i=1}^{n} X(i,1)} \times 100\% \tag{2}$$

The number of misclassified point clouds is defined as Commission Error (*CE*).

$$CE = \frac{\sum_{i=1}^{n} X(i,1) - X(1,1)}{\sum_{i=1}^{n} X(i,1)} \times 100\% = 1 - UA \tag{3}$$

Producer Accuracy (*PA*): The meaning of coordinates listed in Category A in the error matrix represents how much of the category A in this study in LiDAR does belong to genuine category A. The remaining $X(1, 2)$ to $X(1, N)$ actually belong to genuine category B to N. Therefore, the producer accuracy is defined as the point cloud that this study classifies as category A and the ratio of the genuine local category A.

$$PA = \frac{X(1,1)}{\sum_{i=1}^{N} X(1,i)} \times 100\% \tag{4}$$

Except for $X(1, 1)$ point clouds, which are classified as category A in this study and are truly category A, the remaining point clouds are defined as Omission Error (*OE*) because they are missing and are not classified as category A.

$$OE = \frac{\sum_{i=1}^{N} X(1,i) - X(1,1)}{\sum_{i=1}^{N} X(1,i)} \times 100\% = 1 - PA \tag{5}$$

After completing the error matrix, the diagonal elements $X(1, 1)$, $X(2, 2)$, $X(3, 3)$ ..., $X(n, N)$ are the number of point clouds that are fully classified correctly, so the number of point clouds

that are fully classified correctly is defined, and the ratio to all cloud numbers is the overall accuracy (*OA*).

$$OA = \frac{\sum_{i=1}^{N} X(i,i)}{\sum_{i=1}^{n} \sum_{j=1}^{N} X(i,j)} \times 100\% \tag{6}$$

The Kappa statistics, which are generated by the mutual operation of error matrices, can show the error of the whole point-cloud classification, and take into account the factors of misjudgment and omission, so as to provide an indicator of how good the classification results are compared with random classification.

$$Kappa = \frac{Total\ Overall\ Accuracy - Expected\ Accuracy}{1 - Expected\ Accuracy} = \frac{N \sum_{i=0}^{n} X_{ii} - \sum_{i=0}^{n} (X_{i+} \cdot X_{+i})}{N^2 - \sum_{i=0}^{n} (X_{i+} \cdot X_{+i})} \tag{7}$$

## 3. Research Process and Methods

### 3.1. Research Area

This article takes the campus located in Taoyuan city as the study area (as shown in Figure 1). The image is the orthophoto taken by LiDAR during the same period. Though its range is only 650 m × 600 m this area is appropriate to conduct verification of classification results. The average altitude in research area is about 200 m, and the features covered include: different kinds of trees, buildings, asphalt roads, PU ground, cement floor, grassland, and other features. The floor height of buildings in the area is less than three floors, and the grassland is mainly short turf. The collected LiDAR data are the LMS-Q680i full waveform airborne LiDAR data. Considering that the ground features (vegetation) will show different scenes depending on the season, the satellite images collected are mainly WorldView-2 satellite images.

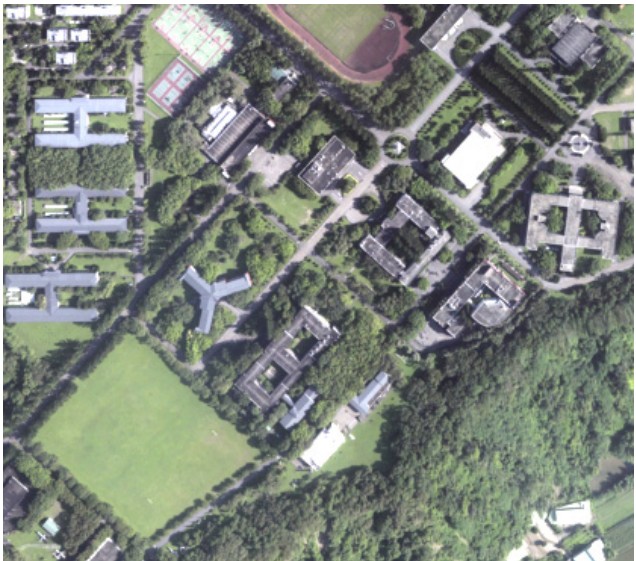

**Figure 1.** Orthophoto of the study area. (650 m × 600 m).

### 3.2. Work Flow

This research focuses on the analysis of LiDAR data and satellite images. We used LiDAR parameters and satellite images to classify objects based on LiDAR point-cloud data and added image parameters to the point-cloud data to verify whether the classification accuracy can be improved. The work flow chart is shown in Figure 2 below.

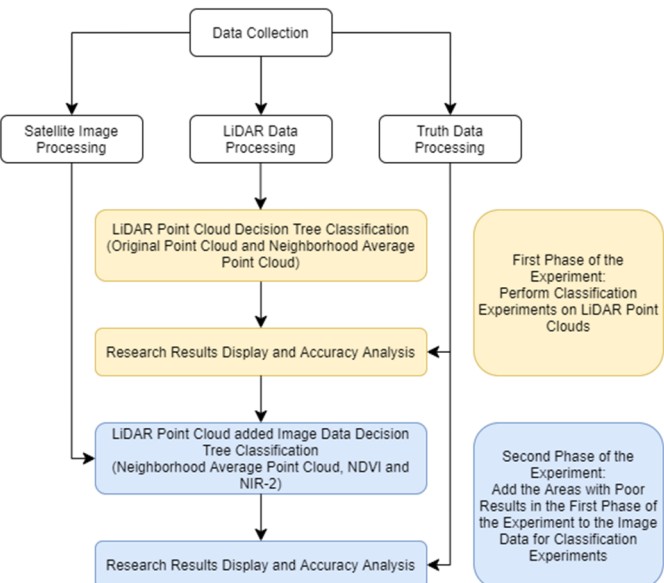

**Figure 2.** Work Flow Chart.

### 3.3. Data Collection and Processing of Satellite Image

#### 3.3.1. WorldView-2 Satellite Imagery

WorldView-2 is the third high-resolution optical satellite owned by American Digital Globe. It has a high optical resolution of 50 cm and a commercial high-resolution optical remote sensing satellite that provides eight bands. The multi-spectral image data of eight bands provide more parameters for remote sensing research. In addition to the traditional multi-spectral bands red, green, blue, and near infrared-1 (NIR-1), four bands such as red-edge, coastal, yellow, and near infrared-2 (NIR-2) are added. The NIR-2 band has a spectral range between 860 and 1040 nm. Compared to NIR-1, this band is less affected by interference from the Earth's atmosphere and is suitable for research and application of plants or biological quality.

#### 3.3.2. Image Fusion and Processing

Image Fusion technology can effectively combine high-resolution full-color images and low-resolution multi-spectral color images to fuse together into a quantity of data that preserves both high-resolution images and multi-spectral color images. The purpose is to obtain more information than a single image. Improving image quality for photointerpretative data fusion is discussed by Gross and Schott [16] and van der Meer [17].

In this study, ERADS IMAGINE software was used to process image fusion, and the method used was Principal Component Analysis (PCA) for computation fusion. In the World View-2 images collected by the institute, the resolution of the multi-spectral image in Figure 3a,b is 2.2 m, and the resolution of the full-color image in Figure 3c,d is 0.5. m. Using ERDAS IMAGINE software, the main components of the two types of images are fused. After the fusion, the resolution of the color image is increased to 0.6 m, as shown in Figure 3e,f.

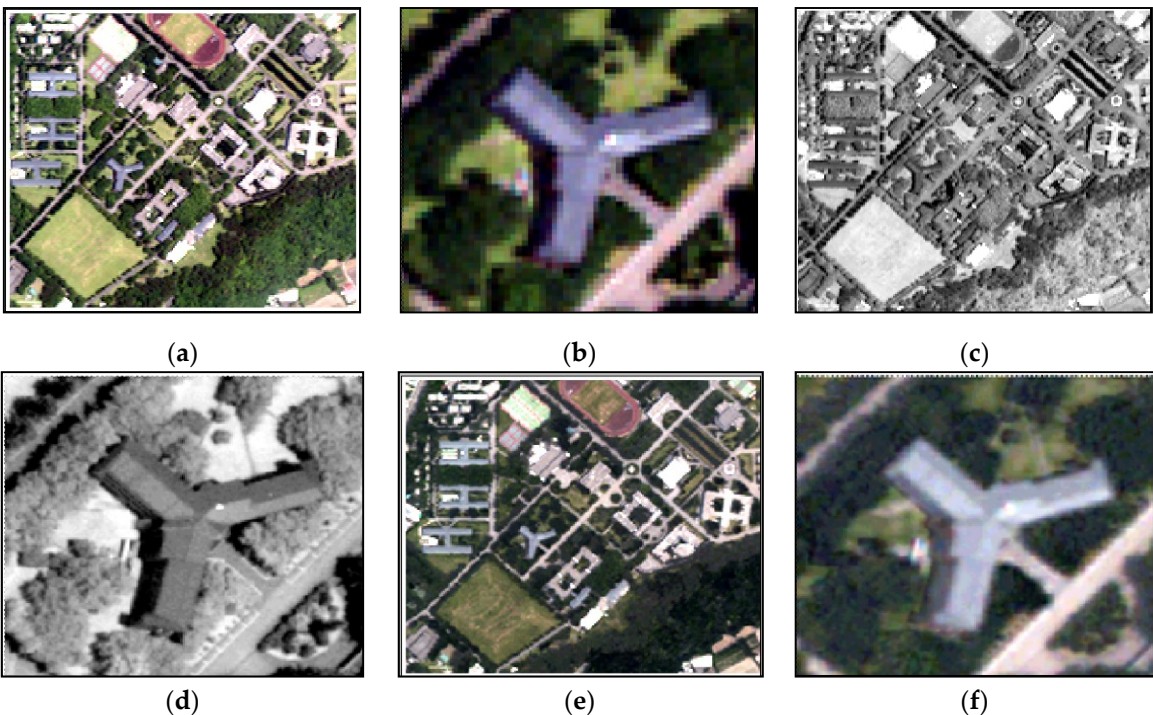

**Figure 3.** Fusion processing. (**a**) Multispectral image; (**b**) Multispectral zoom image; (**c**) Full color image; (**d**) Full color zoom image; (**e**) Image after fusion; (**f**) Fusion zoom image.

### 3.3.3. NDVI Calculation

We use the NIR−1 band and the red band in the satellite image to calculate the NDVI value and use the Matlab software to calculate the NDVI image (as shown in Figure 4).

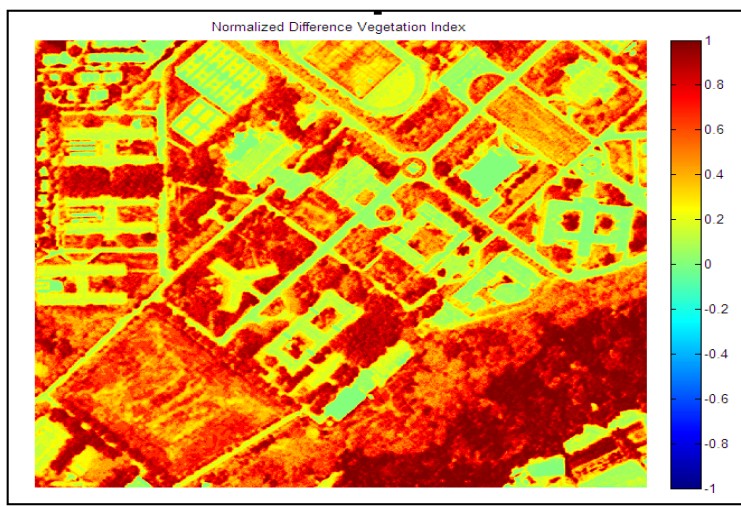

**Figure 4.** NDVI image.

### 3.4. Data Collection and Processing of LiDAR Airborne Image

### 3.4.1. LiDAR Point Cloud Data Collection and Processing

The LiDAR data collected in this article are obtained by scanning the Taoyuan city with LMSQ680i airborne LiDAR. LMS-Q680i is a full waveform no-load LiDAR system manufactured by Riegl Company. It has a good linear scan. The scanning speed of the point arrangement method reaches 266,000 points/s, and the scanning line is 10–200 lines/s. In the study, the positioning and orientation system on the LiDAR and the author's campus armaments building GPS ground tracking station were simultaneously tested. After the

LiDAR data were downloaded, the simultaneous joint measurement ground tracking station data was imported into the LiDAR data for processing. We obtained the correct coordinate point-cloud data, and the relevant information of the positioning and orientation system is shown in Figure 5.

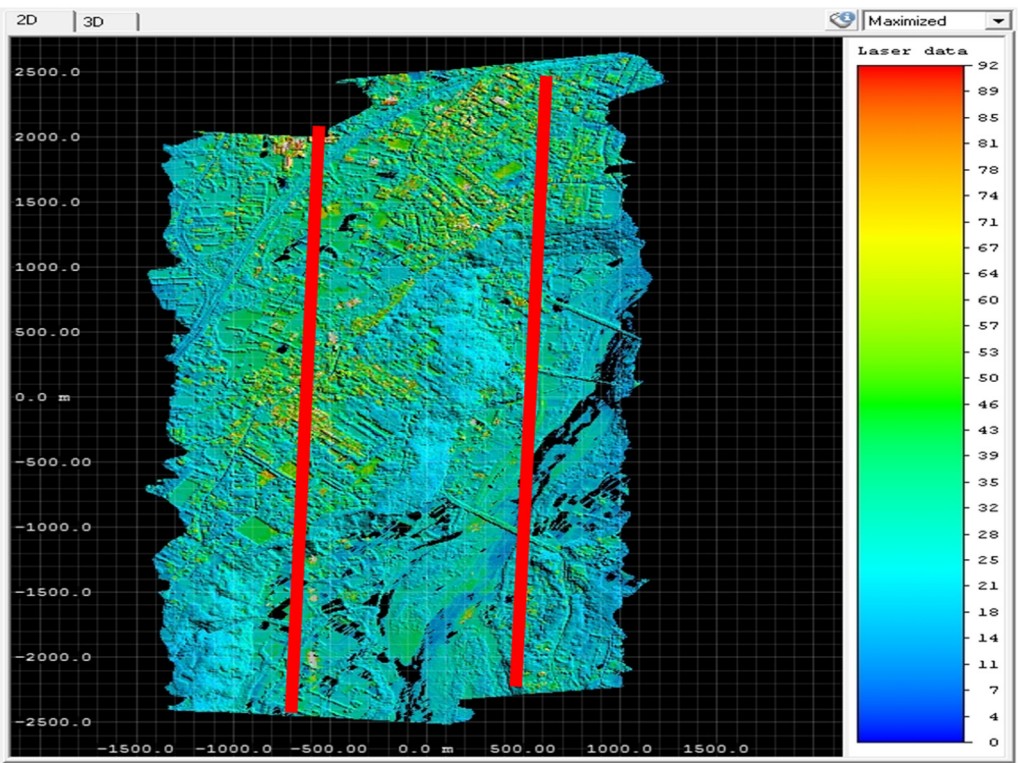

**Figure 5.** Point cloud data of LiDAR's two flight belts.

The LiDAR data used in this study area were taken for the LiDAR data of the two flight bands (as shown in Figure 5). The average measurement height is about 1500 m, and the average point-cloud density is 2 points/m.

Since the data format of each company's LiDAR system is different, in order to allow the data of different companies to be interchangeable, the American Society for Photogrammetry and Remote Sensing (APSRS) issued the "ASPRS LIDAR Data Exchange Format Standard (LAS1.0)" in May 2003. LAS successively released LAS1.1, 1.2, 1.3, 1.4 and 2.0 versions.

This research mainly uses the LAS1.2 data format. In the research, the Riegl program was also used to convert the ASCII file data format and the ASCII output parameters of the pulse width. After obtaining the point-cloud data of the study area, the first stage of the study is to classify and compare the accuracy between the original point-cloud data and the point-cloud data after the neighborhood average.

The point-cloud neighborhood averaging in this research mainly uses the Neighborhood Toolset command in the ArcGIS software neighborhood analysis tool set to perform point-cloud averaging analysis. Uses the width, amplitude, and slope values in the light data to enter the decision-tree classification operation. In the original data, it was found that the distribution of the point-cloud eigenvalues was too discrete, resulting in too much noise, so neighborhood analysis was used to analyze the space. The characteristics of the point cloud are averaged to filter out excessive noise values, which obtains a more average feature distribution. The analysis results are shown in Figures 6–8, where Figure 6a is the image map presented by the original data, and Figure 6b is the image averaged by the neighborhood. It can be seen from the figure that the average color distribution of the image after the neighborhood is smoother, indicating that too much noise has been filtered

out. Too much noise will cause misjudgment of decision-tree classification. The research expects to improve the classification accuracy after filtering to remove noise.

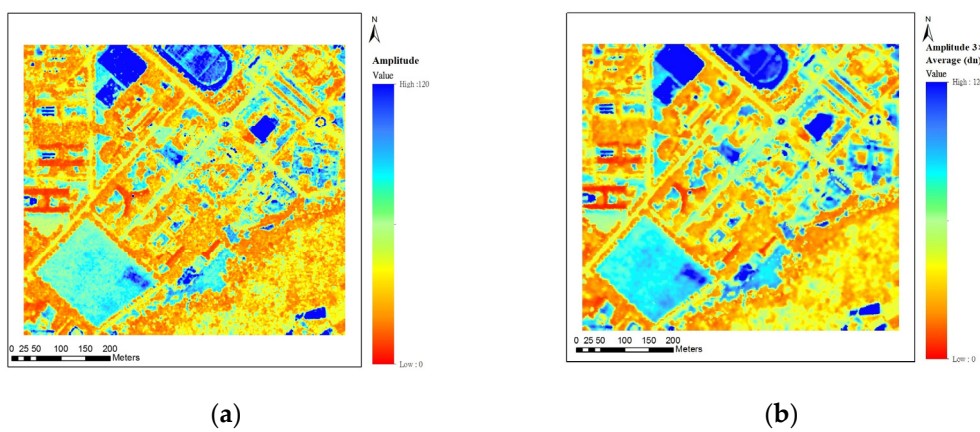

(**a**)                                                      (**b**)

**Figure 6.** Amplitude image. (**a**) Raw amplitude image; (**b**) Amplitude value image after neighborhood average.

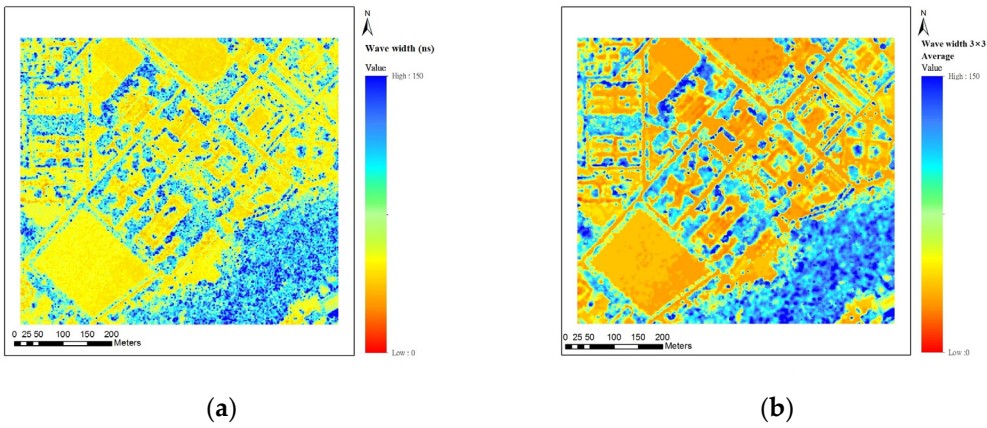

(**a**)                                                      (**b**)

**Figure 7.** Pulse width image. (**a**) Original pulse width image; (**b**) Pulse width value image after neighborhood average.

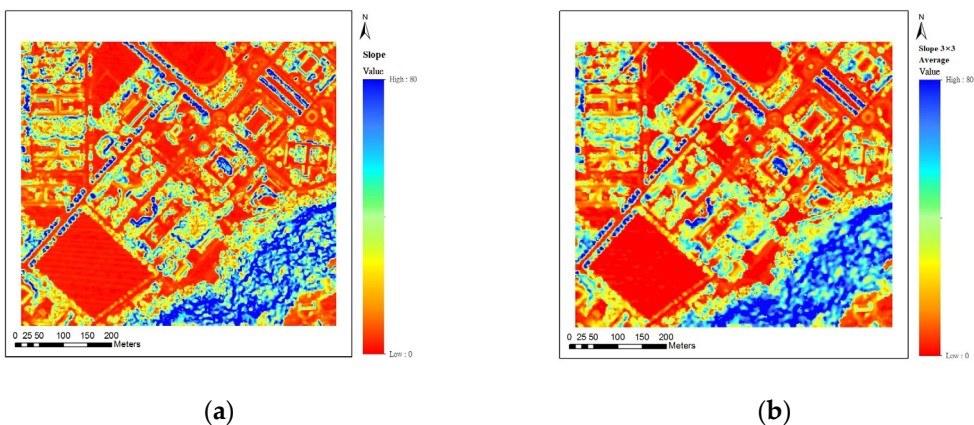

(**a**)                                                      (**b**)

**Figure 8.** Slope value image. (**a**) Original slope value image; (**b**) Slope value image after neighborhood average.

### 3.4.2. LiDAR Data Feature Value Extraction

The LiDAR parameters mainly used in this study include amplitude value, pulse width value, slope value, and custom height value. The amplitude value and wave width

value are the values calculated by Gaussian fitting of the echo waveform by Riegl program. After fitting, the waveform presents a Gaussian distribution curve [18]. The amplitude value is the highest value of the waveform signal intensity, the full width half maximum (FWHM) of the waveform is the wave width value, and FWHM is defined as the wave width value of the waveform when the standard deviation of Gaussian distribution is $2\sqrt{2ln2}\sigma \approx 2.3548\sigma$. The amplitude and slope width of point cloud can be obtained from the waveform fitted by Riegl program [19]. The slope value is the point-cloud slope value calculated by using the slope analysis in ArcGIS software. The slope represents the steepest position on the surface inclination. The LiDAR data are three-dimensional spatial point data. The features in the study area can be classified according to their heights, which can be used as classification parameters. In the study area, according to the actual observation of the terrain by people, most of them are grassland and 0.5 m in height. The ground man-made structures are the main types, 0.5 to 2.5 m are mainly the middle canopy and short buildings, and 2.5 m or more are the high buildings and the tall canopy. Therefore, the study set these three levels as the custom height parameter values. The first layer is 0 to 0.5 m, the second layer is 0.5 to 2.5 m, the third layer is more than 2.5 m (as shown in Figure 9), and the self-defined height value is used as the classification parameter of the decision tree.

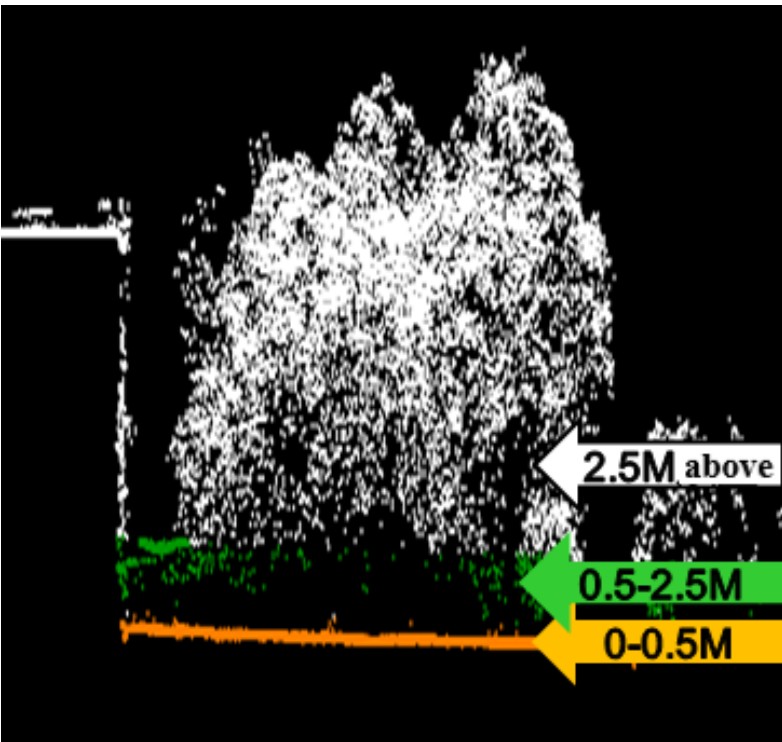

**Figure 9.** Schematic diagram of custom height value.

*3.5. Ground Truth Data Search*

The true value data in this study area are divided into canopy, building, grassland, road, and other ground (PU, cement land, bare mud land) through actual exploration and comparison of aerial photograph data. After the completion of the classifications, the classified images are used as the reference for truth value, and then the true value of the point cloud is plotted. The true value of the point cloud is shown in Figure 10. The truth data are mainly used in the verification of the classification results.

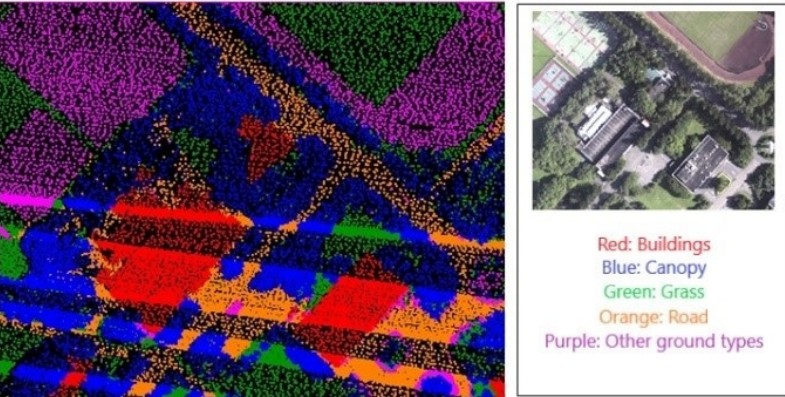

**Figure 10.** Point cloud truth data.

## 4. Results

This study is divided into two experimental stages. The first stage experiment is to compare and analyze the original point cloud with the point cloud after neighborhood average. The second stage experiment reanalyzes the areas with poor classification in the first stage experiment, reduces the study area, and resamples. At the same time, the NDVI and NIR-2 band values in the data and the point-cloud data are added to re-calculate the decision tree. After completion, the results are compared and analyzed the results, and determined whether adding image data can improve the classification accuracy of point clouds.

The pruning levels of the untrimmed classification decision tree are different. In order to have a unified comparison standard, the decision tree is pruned into 10 pruning levels as the basis for accuracy comparison, and then the error and fitting status of the 10 pruning levels are observed through the fitting diagram, so as to judge the characteristics of the classification data. The operation formula of each node of the decision tree is the result judged by the sampling data of the decision-tree operation. The closer the root node operation formula is, the more important its parameters are.

### 4.1. The First Stage Experiment: Original Value and Neighborhood Mean Point Cloud Classification

The first phase of the experiment mainly used LiDAR data for classification. In this paper, the features in the study area are divided into five categories, namely building, canopy, grassland, road, and other ground categories, while other ground categories include cement ground, and PU ground, and bare mud land. The original values of the point cloud and the point cloud averaged by the neighborhood are classified into decision trees, and the accuracy of the results is evaluated after the classification is completed.

First, we picked training samples in the study area and circled the training areas for the five feature categories. The circled training area is formed by overlaying LiDAR point-cloud data with aerial photograph orthophoto images taken at the same time to help understand the point. According to the actual distribution of cloud features, and after the actual feature survey of the researchers, the scope of the training area was selected for the study area, as shown in Figure 11. The colors selected in the training area are: red for buildings, blue for crowns, green for grasslands, orange for roads, and purple for other ground types. Sampling is to sample the point cloud according to the self-defined height of the study. The point-cloud samples of the second and third floors of the self-defined height values are taken for buildings and tree crowns, and the point-cloud samples of the second and first floors are taken for roads, grasslands and other ground classes. There are overlapping areas of blue and purple in the sampling area. Blue is the point cloud on the crown, and purple is the point cloud under the crown. We used the amplitude, width, and slope values in the LiDAR data to draw a histogram of the sampled training area data and recorded the histogram statistics in Tables 2–4. The statistical table shows the ratio of the

average amplitude value of the neighborhood. The overall standard deviation value of the original is reduced by 21.61%, the neighborhood average pulse width value is lower than the original standard value by 55.07%, and the neighborhood average slope value is lower than the original standard value by 25.45%, so this means that the point cloud is averaged in the neighborhood, which can filter out the excessive noise of the point cloud and improve the average distribution of the point cloud's eigenvalues.

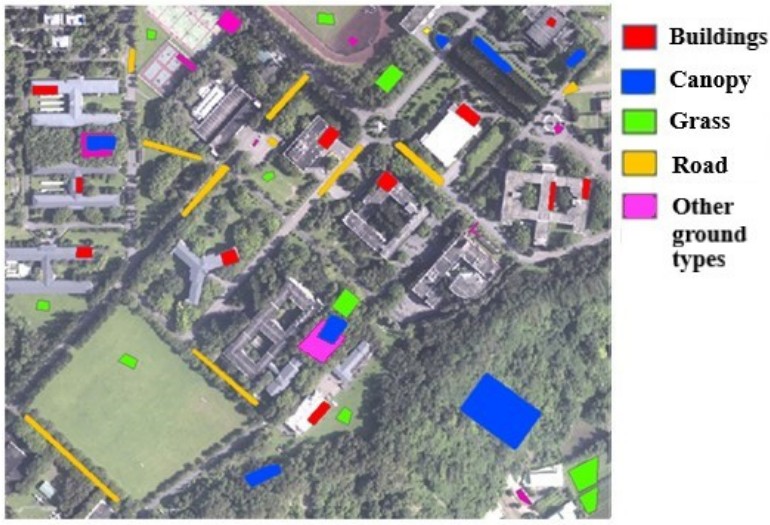

**Figure 11.** Sampling diagram of training samples in the first phase of the experimental study area.

**Table 2.** Training Amplitude Value–Comparison Table of Original Value and Neighborhood Standard Deviation.

| Amplitude Value (Unit: DN) | | | | |
|---|---|---|---|---|
| Method<br><br>Category | Original Mean | Original Value Standard Deviation | Neighborhood Average | Neighborhood Standard Deviation | The Average Standard Deviation of Various Neighborhoods Decreases |
| Buildings | 32.72 | 17.7 | 35.18 | 16.60 | 6.32% |
| Tree | 21.92 | 8.02 | 21.42 | 3.61 | 55.04% |
| Other ground | 2.49 | 3.41 | 1.72 | 2.37 | 30.40% |
| Grass | 32.67 | 13.69 | 32.30 | 11.50 | 16.00% |
| Road | 28.21 | 7.45 | 28.13 | 5.3 | 28.94% |
| Overall standard deviation | | 50.27 | | 39.38 | |
| Neighborhood average overall standard deviation decreases: 21.61% | | | | | |

**Table 3.** Comparison table of training pulse width value–original value and neighborhood mean standard deviation.

| Amplitude Value (Unit: DN) | | | | |
|---|---|---|---|---|
| Method<br><br>Category | Original Mean | Original Value Standard Deviation | Neighborhood Average | Neighborhood Standard Deviation | The Average Standard Deviation of Various Neighborhoods Decreases |
| Buildings | 45.19 | 6.11 | 44.84 | 2.71 | 55.60% |
| Tree | 54.96 | 9.69 | 54.45 | 2.26 | 76.68% |
| Other ground | 43.64 | 2.46 | 43.17 | 1.15 | 53.21% |
| Grass | 47.21 | 7.46 | 46.82 | 5.74 | 23.04% |
| Road | 43.31 | 1.95 | 42.96 | 0.57 | 70.92% |
| Overall standard deviation | | 27.68 | | 12.43 | |
| Neighborhood average overall standard deviation decreases: 55.07% | | | | | |

**Table 4.** Training slope value–original value and neighborhood standard deviation comparison table.

| Method<br><br>Category | Amplitude Value (Unit: DN) | | | | |
|---|---|---|---|---|---|
| | Original Mean | Original Value Standard Deviation | Neighborhood Average | Neighborhood Standard Deviation | The Average Standard Deviation of Various Neighborhoods Decreases |
| Buildings | 16.39 | 17.13 | 15.62 | 12.60 | 26.44% |
| Tree | 33.56 | 15.83 | 33.09 | 11.99 | 24.26% |
| Other ground | 2.49 | 3.41 | 1.723 | 2.37 | 30.40% |
| Grass | 1.22 | 1.45 | 0.74 | 1.19 | 18.16% |
| Road | 1.02 | 0.62 | 0.64 | 0.507 | 17.83% |
| Overall standard deviation | | 38.43 | | 28.65 | |
| Neighborhood average overall standard deviation decreases: 25.45% | | | | | |

### 4.1.1. Decision Tree Classification Based on Original Values

The original amplitude value, wave width value, slope value, and the point-cloud value in the sampling area of self-determined height value are classified by MATLAB. The pruning level of the original decision tree without pruning is 83 levels, and the decision tree is pruned to 10 levels, as shown in Figure 12. After completing the decision-tree calculation, we extract all the LiDAR point clouds in the study area, and extract the point-cloud data that conform to the decision-tree classification operation, as shown in Figure 13. Figure 13a–c respectively represent the results of local point-cloud profile and the results of point-cloud classification in the study area, as well as the comparison and explanation drawings. Figure 13a shows the point-cloud profile area shown in the light blue box of Figure 13c, from which the distribution of point-cloud results can be observed. Figure 13b shows the classification results of the whole study area, and Figure 13c shows the color representative significance in the comparison aerial photos and explanation results.

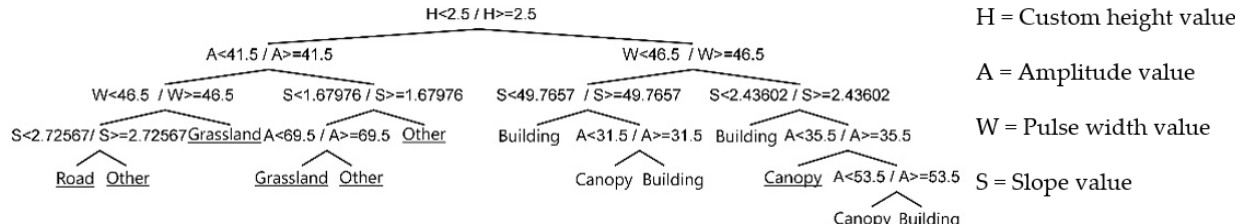

**Figure 12.** Pruning the decision tree with the original value of the first stage experiment.

### 4.1.2. Decision Tree Classification Based on Neighborhood Average

The amplitude value, wave width value, slope value, and self-determined height value point-cloud data in the neighborhood average point-cloud data are classified by the MATLAB program. The pruning level of the original decision tree without pruning is 58 levels, and the branches are pruned to 10 levels, as shown in Figure 14. After completing the decision-tree calculation, we classify and extract all the LiDAR point clouds in the study area and extract the point-cloud data that conform to the decision-tree classification calculation, as shown in Figure 15. Figure 15a–c respectively represent the results of the local point-cloud profile and the results of point-cloud classification in the study area, as well as the comparison and explanation drawings. Figure 15a shows the point-cloud profile area shown in the light blue box of Figure 15c, from which the distribution of point-cloud results can be observed. Figure 15b shows the classification results of the whole study area, and Figure 15c shows the color representative significance in the comparison aerial photos and explanation results.

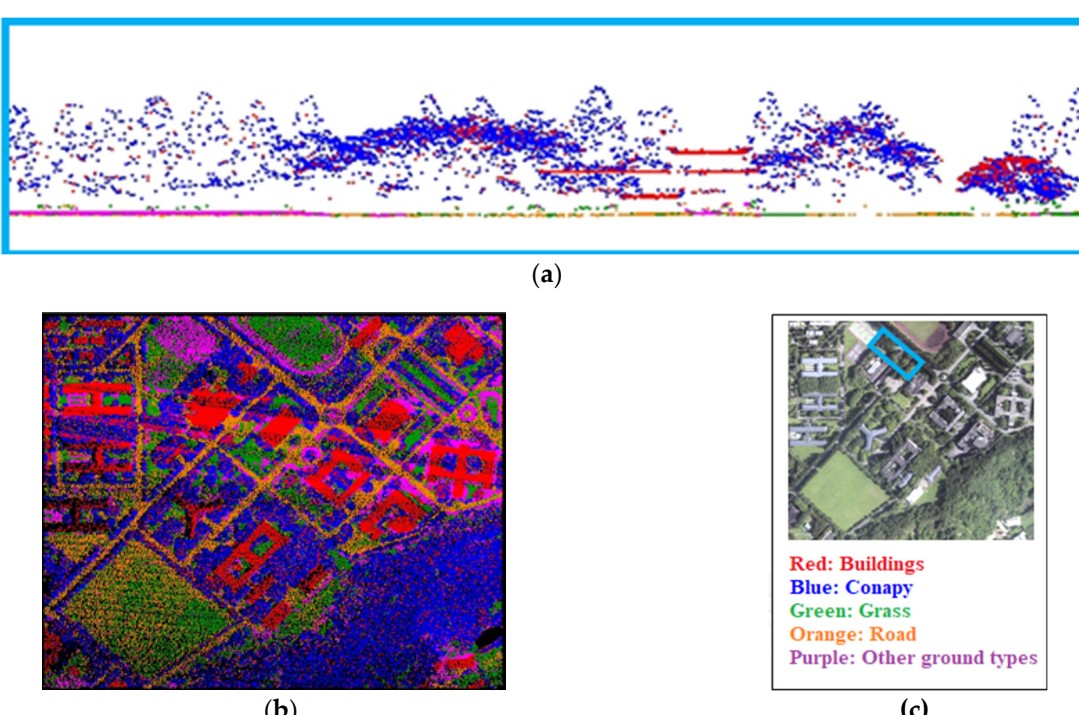

**Figure 13.** Classification results of the original value of the first stage experiment. (**a**) Local point cloud profile results; (**b**) Point cloud classification results in the study area; (**c**) Comparison diagram.

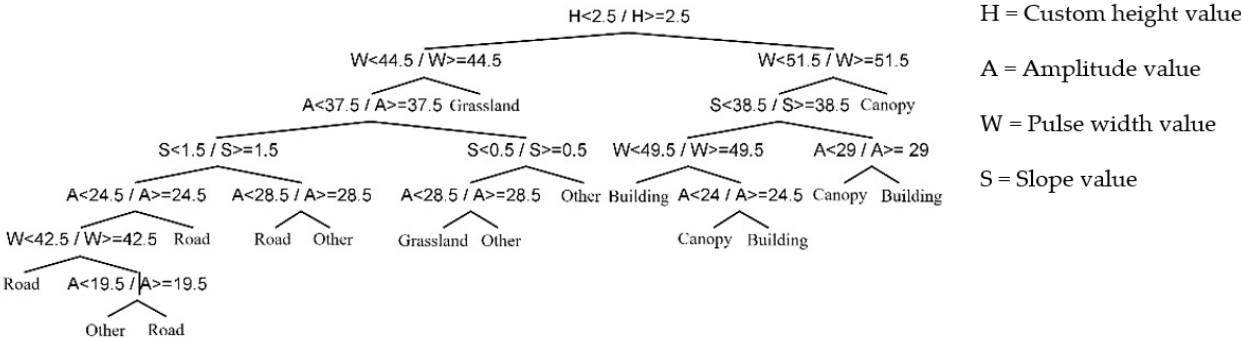

**Figure 14.** Decision tree trimmed by the mean value of neighborhood in the first stage experiment.

### 4.1.3. Research Results Display and Accuracy Analysis

We calculated the error matrix of the truth data and the classified results (as shown in Tables 5 and 6), found the classification accuracy of the original value and the average of the neighborhood, and the overall accuracy of the average of the neighborhood in the selected verification area of 83% was higher than the original value of 7%; Kappa is higher than the original value of 5.92%. The accuracy evaluation results verify that the point-cloud value classification after the neighborhood average can improve the classification results of the original value. From the error matrix, the point-cloud classification can be observed on the ground. The classification accuracy of objects is relatively poor. The average ground category of neighborhoods is 87%, 58% and 48% for roads, grasses and other grounds, respectively. Compared with 96% for buildings and 97% for canopy, the classification accuracy is relatively low.

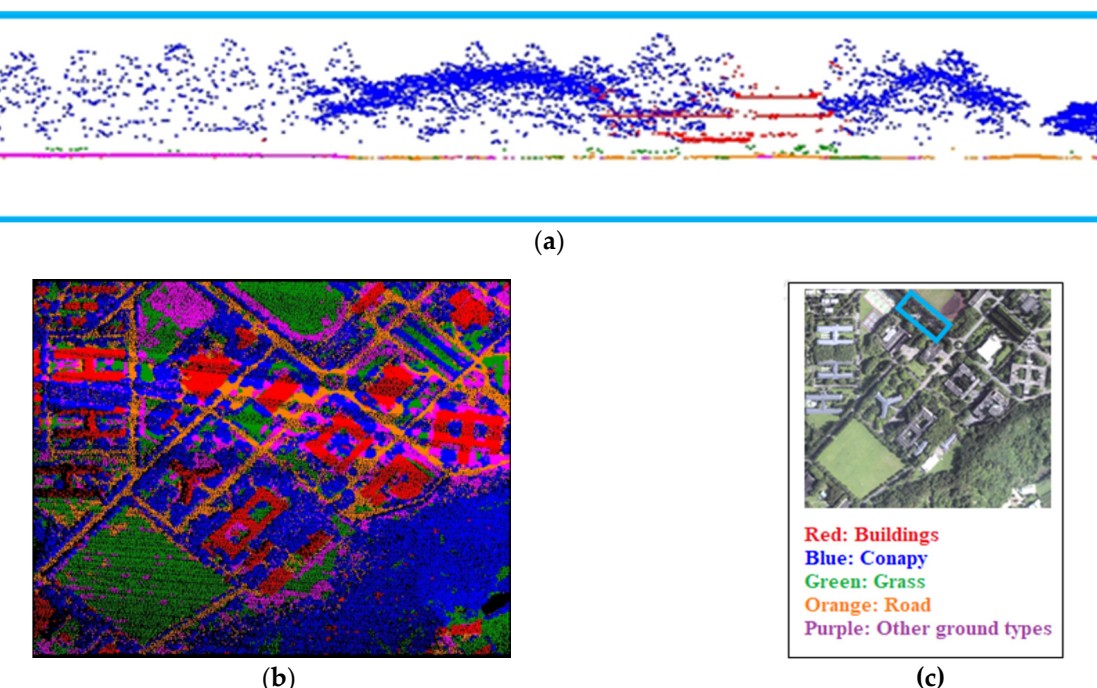

**Figure 15.** Results of the neighborhood average classification in the first phase of the experiment. (**a**) Local point cloud profile results; (**b**) Point cloud classification results in the study area; (**c**) Comparison diagram.

**Table 5.** Error matrix of original value in the first stage experiment.

| Classification Results | Buildings | Canopy | Road | Grassland | Other Ground | Classification Total | Producer Accuracy (PA) |
|---|---|---|---|---|---|---|---|
| Buildings | 22,153 | 6073 | 0 | 0 | 42 | 28,268 | 78% |
| Canopy | 2362 | 38,226 | 0 | 0 | 8 | 40,596 | 94% |
| Road | 0 | 34 | 26,471 | 7081 | 2044 | 35,630 | 74% |
| Grassland | 4 | 419 | 3679 | 10,129 | 7738 | 21,969 | 46% |
| Other ground | 2 | 63 | 2537 | 1659 | 7224 | 11,485 | 63% |
| Ground truth category total | 24,521 | 44,815 | 32,687 | 18,869 | 17,056 | 137,948 | |
| User Accuracy (UA) | 90% | 85% | 81% | 54% | 42% | | |
| Overall Accuracy (OA): 76%, Kappa: 43.26% | | | | | | | |

### 4.2. Second Stage Experiment: Adding Image Data to the Classification of the Neighborhood Average

The results of the first-stage experimental research show that the decision-tree classification averaged in the neighborhood has better results, but the classification results in the ground category (grassland, road, other ground categories) are less satisfactory, so the second stage experiment takes the first in the stage where the experimental classification is not good, and the NDVI value and satellite image NIR-2 band data are added for classification and accuracy evaluation. The question is whether the addition of image information in the experiment can improve the classification accuracy of the point cloud.

**Table 6.** The mean error matrix of the neighborhood in the first stage experiment.

| Classification Results | Buildings | Canopy | Road | Grassland | Other Ground | Classification Total | Producer Accuracy (PA) |
|---|---|---|---|---|---|---|---|
| Buildings | 23,648 | 988 | 0 | 0 | 31 | 24,667 | 96% |
| Canopy | 866 | 43,328 | 0 | 0 | 19 | 44,213 | 98% |
| Road | 0 | 18 | 28,377 | 4536 | 687 | 33,618 | 84% |
| Grassland | 7 | 477 | 1133 | 11,038 | 8070 | 20,725 | 53% |
| Other ground | 0 | 4 | 3177 | 3295 | 8249 | 14,725 | 56% |
| Ground truth category total | 24,521 | 44,815 | 32,687 | 18,869 | 17,056 | 137,948 | |
| User Accuracy (UA) | 96% | 97% | 87% | 58% | 48% | | |
| Overall Accuracy (OA): 83%, Kappa: 49.18% | | | | | | | |

The second phase of the experiment mainly carried out five classifications of LiDAR data, namely building, canopy, grassland, road, and other ground categories, among which other ground categories include cement ground, PU ground, and bare mud land. After that, NDVI and NIR-2 point clouds were added to the neighborhood mean value to classify the decision tree, and the accuracy of the results was evaluated after the classification was completed.

4.2.1. Circle Selection in Decision Tree Training Area

We re-calculated the data of the study area. Since the results of the first-stage experiment classification found that it is easier to misjudge in the ground category, the area with more misjudgments in the study area was taken as the study area of the second-stage experiment (as shown in Figure 16), the NDVI value and the image NIR-2 value were added to the LiDAR data attribute data, and the point cloud was sampled according to the customized height studied during the sampling. The colors selected in the training area are: red for buildings, blue for canopy, green for grassland, orange for roads, and purple for other ground classes. The point-cloud samples of the second and third floors of self-defined height values are taken for buildings and canopy, and the point-cloud samples of the second and first floors are taken for roads, grasslands, and other ground classes.

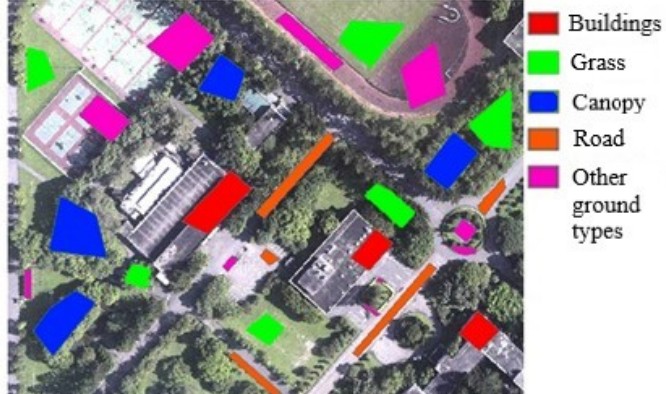

**Figure 16.** Sampling diagram of training samples in the second phase experimental study area.

4.2.2. Decision Tree Classification Based on Neighborhood Average

The amplitude value, wave width value, slope value and self-determined height value point-cloud data in the neighborhood average point-cloud data are classified by the MATLAB program. The pruning level of the original decision tree without pruning is 38 levels, and the branches are pruned to 10 levels, as shown in Figure 17. We extract the point-cloud data that conform to the classification operation of the decision tree, as shown

in Figure 18. Figure 18a–c respectively represent the results of local point-cloud profile and the results of point-cloud classification in the study area, as well as the comparison and explanation drawings. Figure 18a shows the point-cloud profile area shown in the light blue box of Figure 18c, from which the distribution of point-cloud results can be observed. Figure 18b shows the classification results of the whole study area, and Figure 18c shows the color representative significance in the comparison aerial photos and explanation results.

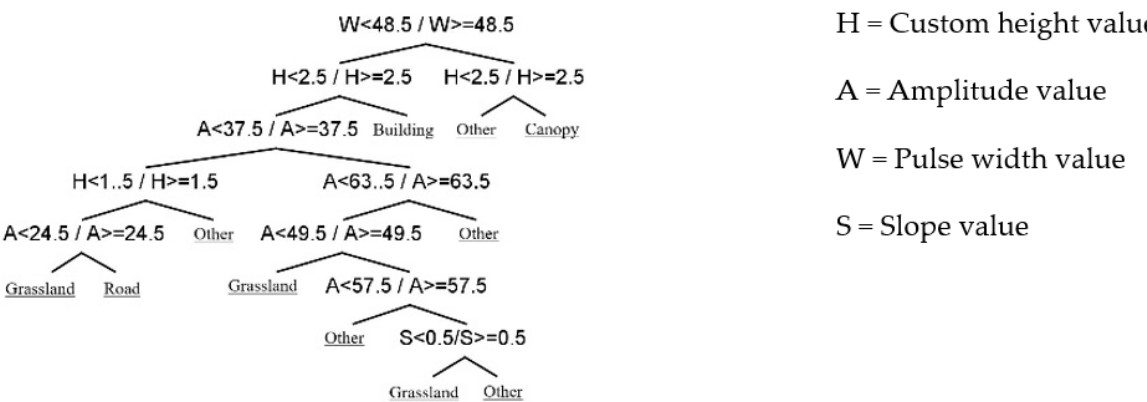

**Figure 17.** Pruning the decision tree by the average value of the experimental neighborhood in the second stage.

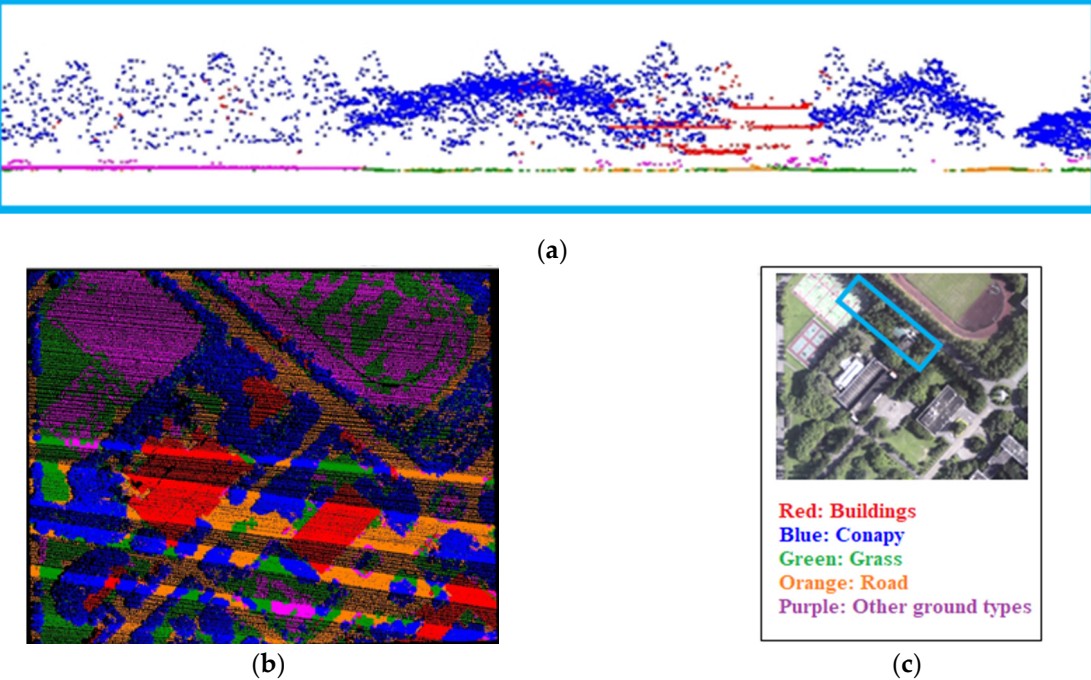

**Figure 18.** Results of the neighborhood average classification in the second experiment. (**a**) Local point cloud profile results; (**b**) Point cloud classification results in the study area; (**c**) Comparison diagram.

4.2.3. Decision Tree Classification Based on the Neighborhood Average Value and NDVI Value

The amplitude value, wave width value, slope value, self-determined height value and NDVI value in the neighborhood average point-cloud data are classified by the MATLAB program. The pruning level of the original decision tree without pruning is 48 levels, and the branches are pruned to 10 levels, as shown in Figure 19. After completing the decision-tree calculation, we extract all the LiDAR point clouds in the study area and extract the point-cloud data that conform to the decision-tree classification operation, as shown in

Figure 20. Figure 20a–c respectively represent the results of local point-cloud profile and the results of point-cloud classification in the study area, as well as the comparison and explanation drawings. Figure 20a shows the point-cloud profile area shown in the light blue box of Figure 20c, from which the distribution of point-cloud results can be observed. Figure 20b shows the classification results of the whole study area, and Figure 20c shows the color representative significance in the comparison aerial photos and explanation results.

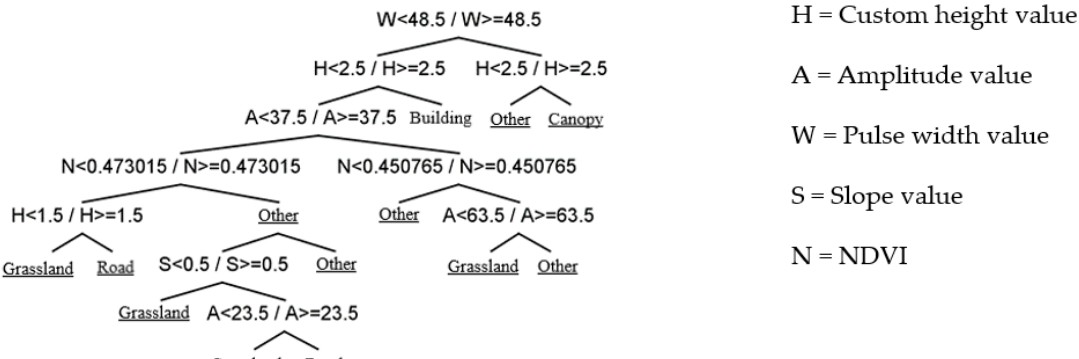

**Figure 19.** The second stage experiment adds NDVI value to trim the decision tree.

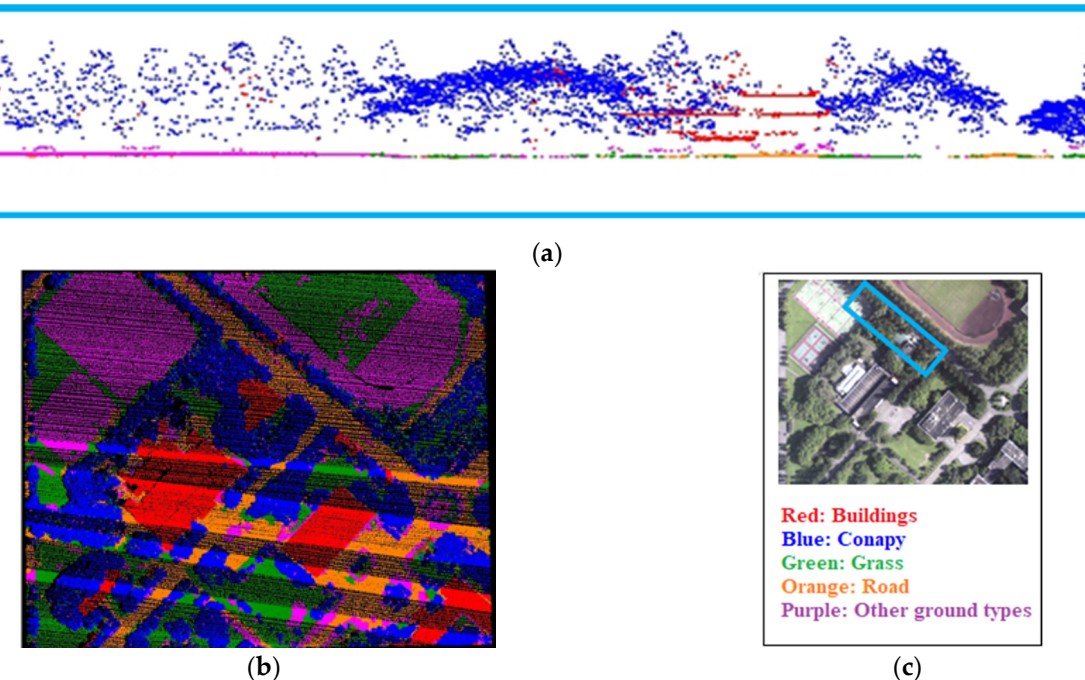

**Figure 20.** The second stage experiment added NDVI value classification results. (**a**) Local point cloud profile results; (**b**) Point cloud classification results in the study area; (**c**) Comparison diagram.

4.2.4. Adding NIR-2 Band Value Decision Tree Classification by Neighborhood Average

The amplitude value, wave width value, slope value, self-determined height value and NIR-2 value in the neighborhood average point-cloud data are classified by the MATLAB program. The pruning level of the original decision tree without pruning is 43 levels, and the branches are pruned to 10 levels, as shown in Figure 21. After completing the decision-tree calculation, we extract all the LiDAR point clouds in the study area, and extract the point-cloud data that conform to the decision-tree classification operation, as shown in Figure 22. Figure 22a–c respectively represent the results of local point-cloud profile and the results of point-cloud classification in the study area, as well as the comparison and explanation drawings. Figure 22a shows the point-cloud profile area shown in the light

blue box of Figure 22c, from which the distribution of point-cloud results can be observed. Figure 22b shows the classification results of the whole study area, and Figure 22c shows the color representative significance in the comparison aerial photos and explanation results.

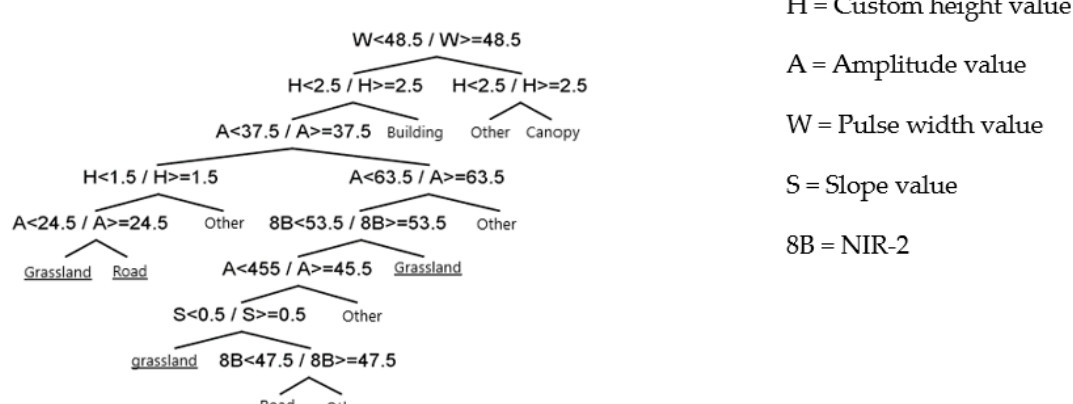

H = Custom height value

A = Amplitude value

W = Pulse width value

S = Slope value

8B = NIR-2

**Figure 21.** The second stage experiment adds image NIR-2 value to trim the decision tree.

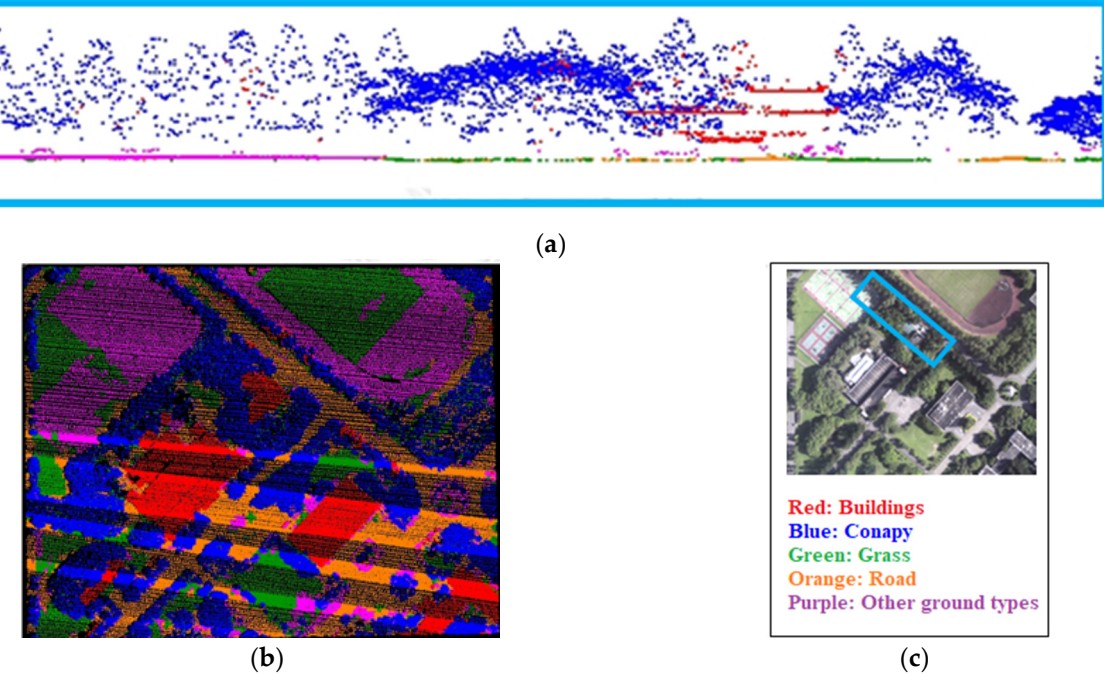

(a)

(b)

(c)

Red: Buildings
Blue: Conapy
Green: Grass
Orange: Road
Purple: Other ground types

**Figure 22.** The second stage experiment added image NIR-2 value classification results. (**a**) Local point cloud profile results; (**b**) Point cloud classification results in the study area; (**c**) Comparison diagram.

4.2.5. Achievement Display and Accuracy Analysis

We calculated the error matrix of the truth data and the classified results (as shown in Tables 7–9), found the neighborhood average, then added the neighborhood average to NDVI and NIR-2 classification accuracy, using the error matrix to compare the differences between its three accuracies.

**Table 7.** Neighbor mean error matrix of the second stage experiment.

| Classification Results | Buildings | Canopy | Road | Grassland | Other Ground | Classification Total | Producer Accuracy (PA) |
|---|---|---|---|---|---|---|---|
| Buildings | 24,029 | 1355 | 0 | 0 | 31 | 25,415 | 95% |
| Canopy | 485 | 42,961 | 0 | 0 | 19 | 43,465 | 99% |
| Road | 0 | 12 | 28,395 | 2965 | 1055 | 32,427 | 88% |
| Grassland | 2 | 0 | 3985 | 13,464 | 4712 | 22,163 | 61% |
| Other ground | 5 | 487 | 339 | 2440 | 11,239 | 14,510 | 77% |
| Ground truth category total | 24,521 | 44,815 | 32,687 | 18,869 | 17,056 | 137,948 | |
| User Accuracy (UA) | 98% | 96% | 87% | 71% | 66% | | |
| Overall Accuracy (OA): 87%, Kappa: 52.60% | | | | | | | |

**Table 8.** Error matrix of neighborhood mean value plus NDVI value in the second stage experiment.

| Classification Results | Buildings | Canopy | Road | Grassland | Other Ground | Classification Total | Producer Accuracy (PA) |
|---|---|---|---|---|---|---|---|
| Buildings | 24,029 | 1355 | 0 | 0 | 30 | 25,414 | 95% |
| Canopy | 485 | 42,961 | 0 | 0 | 19 | 43,465 | 99% |
| Road | 0 | 12 | 27,555 | 1001 | 591 | 29,159 | 94% |
| Grassland | 2 | 0 | 3185 | 16,345 | 651 | 20,183 | 81% |
| Other ground | 5 | 487 | 1947 | 1523 | 15,765 | 19,727 | 80% |
| Ground truth category total | 24,521 | 44,815 | 32,687 | 18,869 | 17,056 | 137,948 | |
| User Accuracy (UA) | 98% | 96% | 84% | 87% | 92% | | |
| Overall Accuracy (OA): 92%, Kappa: 56.85% | | | | | | | |

**Table 9.** Neighbor mean value plus NIR-2 value error matrix in the second stage experiment.

| Classification Results | Buildings | Canopy | Road | Grassland | Other Ground | Classification Total | Producer Accuracy (PA) |
|---|---|---|---|---|---|---|---|
| Buildings | 24,029 | 1355 | 0 | 0 | 31 | 25,415 | 95% |
| Canopy | 485 | 42,961 | 0 | 0 | 19 | 43,465 | 99% |
| Road | 0 | 12 | 28,874 | 4200 | 1365 | 34,451 | 84% |
| Grassland | 2 | 0 | 2677 | 12,514 | 1725 | 16,918 | 74% |
| Other ground | 5 | 47 | 1136 | 2155 | 13,916 | 17,259 | 81% |
| Ground truth category total | 24,521 | 44,815 | 32,687 | 18,869 | 17,056 | 137,948 | |
| User Accuracy (UA) | 98% | 96% | 88% | 66% | 82% | | |
| Overall Accuracy (OA): 89%, Kappa: 53.81% | | | | | | | |

Compared with the average value of the NDVI added to the neighborhood, the overall accuracy of the NDVI value increased by 5%, the Kappa value increased by 4.25%, other ground and grass types in the UA increased by 26% and 16%, and the road type decreased by 3% In the PA part, other ground categories, grass categories and road categories were increased by 9%, 20% and 6%, respectively. The result shows that the ground category classification has become better.

Compared with the NIR-2 single band image value, the nir-2 single band image value improves the overall accuracy by 2%, and the Kappa value increases by 1.21%. In UA, other ground and Road classes increased by 16% and 1%, respectively, and in grassland class

decreased by 5%. In PA, other ground classes and grassland classes increased by 10% and 13%, respectively, and Road classes decreased by 3%. The results show that the distribution of NIR-2 single band image values is better in grassland and other ground classes, so it is proved that adding image information to point-cloud data can effectively improve the classification accuracy of the point cloud. Because the decision tree takes 10 pruning levels in classification, and the decision leaf nodes of building in progress and crown are the same, the classification accuracy of building and crown is the same.

## 5. Conclusions and Suggestions

### 5.1. Conclusions

In this study, the parameters of the full waveform light were used for the classification and accuracy evaluation of the ground features. After verification, the averaged point-cloud results had better classification accuracy, but the classification results were poorer on grasslands and other artificial structures. If it is supplemented with image parameters, the poor part of the point-cloud classification can be improved.

The first phase of the experiment mainly classifies and evaluates the original point cloud and the point cloud averaged by the neighborhood. Through decision-tree classification, the following conclusions are drawn after analysis and research:

(1) The point-cloud data are averaged by neighborhood analysis, and the standard deviation of each parameter value is reduced, which means that the point cloud can effectively filter out noise after neighborhood average, especially for the bandwidth value. In other words, there is the best average distribution.

(2) Comparison was made of the accuracy of the point-cloud decision-tree division between the original point cloud and the neighborhood average. The overall accuracy of the point-cloud classification after neighborhood averaging was 83% higher than the original value of 7%, and Kappa was higher than the original value of 5.92%, so the average of the point-cloud neighborhood can filter out excessive noise and improve the classification accuracy.

(3) Since the neighborhood average converts the point cloud into grid data, and then averages the point clouds in the grid data in a rectangular manner, the overlapping of the point clouds with different values will result in point clouds with different values. After averaging the neighborhood, the average becomes the same value. As shown in Figure 23, the vertically overlapping parts will average the parameter values of buildings and crowns to the same value during averaging, as shown in the yellow box of Figure 23a. After averaging, the entire vertical surface is averaged, so that the building parameter values are averaged to become the canopy class, and the neighborhood average will cause the point cloud to misjudge where different types of features overlap. The neighborhood average point cloud is suitable for use in an open surface area without overlap; if it is used in an area with much shadow, it will cause a false judgment of the overlapping part of the shadow.

In the second phase of the experiment, the areas with poor classification results in the first phase of the experimental research were taken, the scope of the study area was narrowed, the second decision-tree classification was conducted, and re-sampling was conducted. After analysis and research, the following conclusions were drawn:

(1) After averaging the neighborhood point-cloud value for the decision tree operation, the image parameter NDVI value and image NIR-2 value were added, respectively, for the decision-tree operation. Research results show that adding NDVI and NIR-2 can improve the overall accuracy; especially for the ground object categories that are more difficult to distinguish in the first phase of the experiment there are better classification results. The overall accuracy of the NDVI value is increased by 5%, the Kappa value is increased by 4.25%, and the NIR-2 value is added to increase the overall accuracy by 2%. The Kappa value is increased by 1.21%, so LiDAR data combined with satellite image data can effectively improve the classification accuracy of LiDAR point clouds.

(2) Because the difference between the shooting time of the satellite image and the LiDAR image is 1 year, the vegetation has different extent, which will cause errors in the classification. For example, in as shown in Figure 24, the turf part of the playground in the satellite image is bare soil, and the turf on the aerial photo image taken by LiDAR at the same time is the part without bare soil. Therefore, when the decision tree is added to the satellite image parameters for classification, the playground turf will be classified incorrectly.

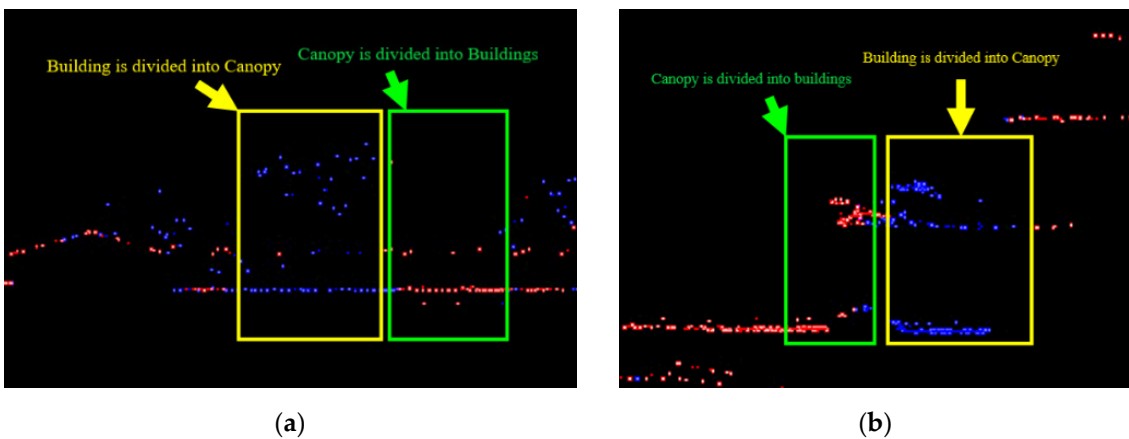

(**a**)                    (**b**)

**Figure 23.** (**a**) Neighborhood average limit graph I; (**b**) Neighborhood average limit graph II.

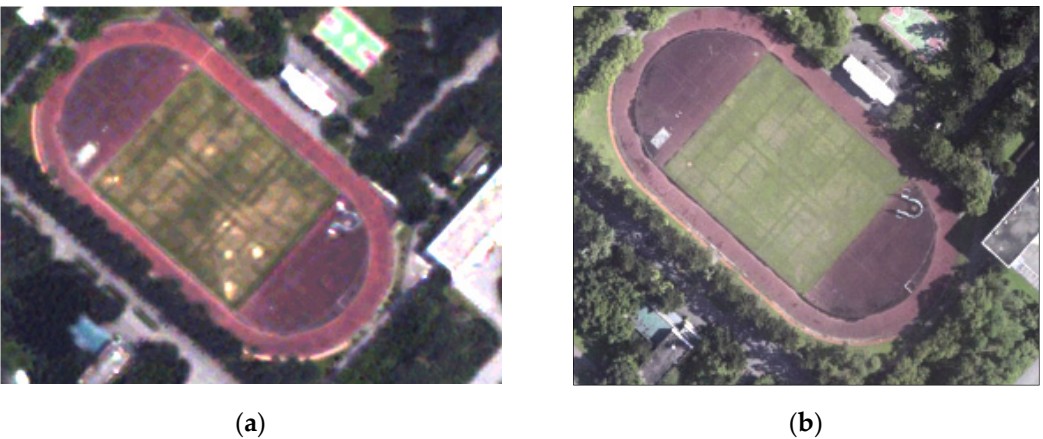

(**a**)                    (**b**)

**Figure 24.** Vegetation Difference Image. (**a**) Satellite imagery; (**b**) Aerial photograph.

*5.2. Suggest*

After this study, the following suggestions are summarized:

(1) Since this research mainly hopes to obtain the classification results of point clouds simply and quickly and provide the application of disaster prevention and relief layers, the selected LiDAR parameters are the parameter data (amplitude value, pulse width value, and slope value) that can be obtained quickly. The classification accuracy of the LiDAR parameters selected in the study is poor in the ground class; it is recommended that other LiDAR parameters (number of echoes, back reflection, elevation difference, multiple echo percentage, etc.) be added for classification research to improve the ground features classification accuracy.

(2) In this study, the neighborhood average point cloud is performed on the entire study area. In order to avoid the misjudgment of overlapping point clouds caused by the neighborhood average, it is recommended that the neighborhood average point cloud can be performed only on the open and uncovered areas to avoid misjudgment. The situation arises, and the classification accuracy can also be improved.

(3) This study uses the CART decision-tree algorithm, which is mainly due to the short-comings of the decision tree without information hiding, and the analysis results can be presented in the form of a decision tree, which can directly see the important changes and their segmentation effects, which can make the research results more clear and understandable. On the other hand, the decision tree is simple and fast. If it is used in image interpretation, it can reduce the labor and time of the interpreter at work and can understand the correlation of the measurement area in a short time. Geographic information provides various scientific and practical applications. At present, many full-waveform classification researches use Random Forest, Support Vector Machine (SVM), Neural Network (NN) and other classification methods for classification. It is recommended that different algorithms can be used for classification to achieve better classification results.

(4) In terms of obtaining LiDAR and images, it is recommended to use LiDAR point-cloud data and satellite images in the same year and season to reduce the differences in ground features and vegetation and avoid lowering the classification accuracy.

(5) The decision-tree classification results in the study are only applicable in this study area. This study has not been verified and compared in different study areas. It is recommended that LiDAR data at different altitudes can be used for classification studies to compare between the different flight zones. The parameter difference of the research, research and analysis of the general LiDAR classification parameter data, to improve the versatility of point-cloud data in different research areas.

(6) This study mainly studies the Red, NIR-1 and near-NIR-2 band information in the World View-2 satellite image, using NDVI and NIR-2 parameters and adding LiDAR point-cloud data for classification. For the advantages and disadvantages and limitations of image bands, it is recommended that each band image be analyzed and studied. After calculating the best classification value between bands, it can be added to the LiDAR point cloud for classification to effectively improve the classification results and accuracy.

**Author Contributions:** Conceptualization, Ming-Da Tsai; Data curation, Chun-Ta Wei; Formal analysis, Yu-Lung Chang; Investigation, Yu-Lung Chang; Methodology, Ming-Da Tsai and Yu-Lung Chang; Project administration, Chun-Ta Wei and Ming-Da Tsai; Resources, Ming-Da Tsai; Software, Ming-Da Tsai; Supervision, Ming-Da Tsai; Validation, Chun-Ta Wei and Yu-Lung Chang; Visualization, Chun-Ta Wei; Writing—original draft, Chun-Ta Wei and Yu-Lung Chang; Writing—review & editing, Chun-Ta Wei and Ming-Chih Jason Wang. All authors have read and agreed to the published version of the manuscript.

**Funding:** This research received no external funding.

**Acknowledgments:** Thanks to Strong Engineering Consulting Co., Ltd. for assisting LiDAR data collection and data preprocessing and other related technical support operations.

**Conflicts of Interest:** The authors declare no conflict of interest.

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
