# Peer review of "Enhancing the Accuracy of Land Cover Classification by Airborne LiDAR Data and WorldView-2 Satellite Imagery"

_ijgi, doi:10.3390/ijgi11070391_

Round 1

Reviewer 1 Report

The topic of this paper is interesting, but I would expect a standard validation framework in order to accept a publication in a scientific journal. Olofsson's paper or Congalton's book are good reference for that. Please use probabilistic sampling so that the quality of the results can be rigourously assessed. I will continue the review with great interest once this has been done. Another point of concern is the lack of clarity of the objectives. Does LIDAR enhance remote sensing or remote sensing enhance LIDAR ? If LIDAR FWF is so popular, then there are probably more papers on the topics than the few cited here, and you should find more application of the combined use. The introduction must be more than the summary of selected papers.

Reviewer 2 Report

This article mainly use the ALS and Satellite image to improve land cover classification accuracy. The combination is interesting and has a potential for publication. However, there are some major concerns after reading this manuscript. Necessary revision is needed in order to publish in the journal. Mayor issues from my view are following:

1 There is two much background introduction. Authors should highlight what is missing for previous research and what is the aim for this research.

2 The theory part can be simplified to one or two paragraphs since lidar, ndvi, decision tree, error matrix, etc are commonly used.

3 I don’t expect too much data processing since the study area is small for ALS and satellite. I highly suggest the authors expand the study region and verify their combination method.

4 in the result part, authors should highlight the difference whether using ALS or not for your classification results.

5 Number of figures are too many for a journal. Figures should focus on your study area, method, very important results that you want to present.

Reviewer 3 Report

See in main document. The article should be totally revised and the different sections (Introduction, Materials and Methods, Results, Discussion and Conclusion) must be respected.

Author Response

The article has been corrected according to your documentation.
One point to explain to you that WorldView-2 satellite imagery is data we purchased ourselves and is not open access.

Round 2

Reviewer 1 Report

I am not satisfied ith the answers to my comments. For a second review I find the overall quality of the presentation unacceptable. For example:

-The introduction still only consists in the summary of 2 (relatively old) scientific papers which are not related in a way to clearly define the state of the art and hence highlight the novelty of the paper. The first paper cited is about random forest, then decision tree is used, which is more simple, without a real justification. Paper 2 (of 2012) obtained a better OA than your paper, maybe not exactly on the same objects, but it would be worth comparing with their method.

-Figures are too small and their caption is sometimes incorrect (e.g. figure 4 states point cloud truth data, but is is NDVI).

- Some technical details are missing or wrong, for example, in the introduction :

"The research will use the image data of the Near-Infrared
band-2 (NIR-2) and Near-Infrared band-1 (NIR-1) to calculate the Nor-malized Difference Vegetation Index (NDVI)"

don't you use the red channel as explained in 3.3.3 ? If so, do you take the average od NIR-1 and 2 ?

-The theory part is unnecessarily long as it describes

- The author state in their answer that they used probabilistic sampling for the validation, but in the manuscript it is still written that the performance comparison is made on a subset of the area, and the figure seems to show that a zone was taken as input.

Author Response

Thanks for your response, please see the attachment.

Reviewer 2 Report

Thanks for considering my comments in the revision.

Author Response

Thanks for your comments.